# The Evolution of Humanitarian Aid in Disasters: Ethical Implications and Future Challenges

**Pedro Arcos González** * and **Rick Kye Gan**

Unit for Research in Emergency and Disasters, University of Oviedo, 33006 Oviedo, Spain; ganrick@uniovi.es
* Correspondence: arcos@uniovi.es

**Abstract:** Ethical dilemmas affect several essential elements of humanitarian aid, such as the adequate selection of crises to which to provide aid and a selection of beneficiaries based on needs and not political or geostrategic criteria. Other challenges encompass maintaining neutrality against aggressors, deciding whether to collaborate with governments that violate human rights, and managing the allocation and prioritization of limited resources. Additionally, issues arise concerning the safety and protection of aid recipients, the need for cultural and political sensitivity, and recognition of the importance of local knowledge, skills, and capacity. The appropriateness, sustainability, and long-term impact of actions; security risks for aid personnel; and the need for transparency and accountability are also crucial. Furthermore, humanitarian workers face the duty to report and engage in civil activism in response to human rights violations and the erosion of respect for international humanitarian law. Lastly, the rights of affected groups and local communities in the decision-making and implementation of humanitarian aid are vital. The traditional foundations and approaches of humanitarian aid appear insufficient in today's landscape of disasters and crises, which are increasingly complex and divergent, marked by a diminished capacity and shifting roles of various actors in alleviating suffering. This article reviews the historical evolution of the conceptualization of humanitarian aid and addresses some of its ethical challenges and dilemmas.

**Keywords:** ethics; disasters; humanitarian aid

## 1. Introduction

Humanitarian aid or humanitarian assistance has traditionally been understood as a set of actions carried out by individuals or institutions addressed at providing assistance to people and communities affected by crises, conflicts, or disasters. It is intended to save lives, alleviate suffering, and maintain human dignity during and after crises and disasters, as well as to prevent and strengthen preparedness for when such situations occur [1]. Humanitarian aid should follow the classical humanitarian principles of humanity, impartiality, neutrality, and independence, rooted in international humanitarian law (IHL), adopted by the United Nations through General Assembly Resolutions 46/182 and 58/114 [2] and incorporated into the Code of Conduct for the International Red Cross and Red Crescent Movement and Non-Governmental Organizations in Disaster Relief [3], as well as the Core Humanitarian Standard on Quality and Accountability [4].

However, as we will examine in this text, the concept of humanitarian aid has evolved, expanding both in scope and intervention approaches. Yet, despite these advancements, it continues to pose significant challenges and dilemmas of ethical nature derived from its substantial core, that is, the human moral duty to help those who experience suffering and have their human rights and survival threatened. In addition to these ethical challenges and dilemmas, humanitarian aid—especially when provided in complex emergency situations—faces political limitations that are typically present in these types of crises. These limitations condition both the aid intervention itself and its outcomes.

## 2. Complex Humanitarian Emergencies

The context in which humanitarian aid operates is within crises and disasters that affect human populations. While a universally accepted definition of a crisis remains elusive, there exists a widely recognized definition of a disaster, as proposed by the United Nations Strategy for Disaster Risk Reduction, as "a serious interruption to the functioning of a society that involves widespread human, material, economic or environmental losses and impacts that exceed the capacity of the affected community to cope with its own resources" [5]. Both crises and disasters are phenomena of global occurrence and distribution with increasing impacts [6]. Neither crises nor disasters are essentially natural phenomena but, to a large extent, the result of elements of a social nature, and their impacts result from patterns of socioeconomic vulnerability, geographic exposure to unmitigated hazards, and inadequate preparation [7–9].

Crises and disasters, from the perspective of social phenomena, are inherently multifaceted, and their conceptualization has evolved significantly over recent decades [10]. In essence, crises and disasters arise from diverse hazards, including natural, technological, and complex emergencies, and differ in their intensity, patterns, and onset speed, consequently shaping the methodological approaches used by researchers [11]. The ethical dimensions of these phenomena, and thus of the humanitarian aid provided, vary distinctly across each type of crisis or disaster.

Furthermore, the United Nations' International Strategy for Disaster Reduction (ISDR) since 2000 and the Sendai Framework's [12] advocacy for a strategy centered on reducing and managing risk and vulnerability represent a paradigm shift and introduce new challenges in disaster research, particularly in research on the ethical elements of disasters.

Among the types of existing man-made disasters, *complex emergencies or complex humanitarian emergencies* pose the greatest challenges and ethical dilemmas due to their component of violence and impacts on life and human rights. Complex emergencies are multifaceted humanitarian crises in a country, region, or society where there is a total or considerable breakdown of authority resulting from internal or external conflict and which usually requires a multi-sectoral, international response. Complex emergencies affect civil populations confronted with generalized violence, associated with forced population displacement and nutritional insecurity, leading to a general degradation of health conditions and producing a dramatic rise in mortality and serious violations of human rights [13,14]. Table 1 presents a categorization of complex emergencies as classified by L. Macias [15] and Figure 1 shows various ongoing complex emergencies in 2023, as detailed in the International Rescue Committee Report [16].

**Table 1.** Typology of complex emergencies (CEs) (source: Macias 2013).

| Main Types of Ces | Components | Examples | Relative Impact Assessment | Possible Responses |
|---|---|---|---|---|
| Type 1 *Acute* | - Acute high-intensity conflict: the level is higher than the country's baseline of violent events<br>- Acute environmental disaster<br>- High level of poverty<br>- Complex social and ethnic geography | **Sudan Nigeria** | - Large affected area<br>- Food insecurity: price hikes<br>- High mortality rates<br>- Concentrated forms of conflict-induced displacement: refugees and internally displaced person (IDP) settlements<br>- Epidemic outbreaks | **Food aid**<br>- Short-term distribution for displaced persons<br>- Protection of refugees and IDPs<br>**Negotiation and coordination**<br>- Open negotiation of humanitarian access with all conflict actors<br>- High coordination between NGOs and agencies<br>- Build resilience |

**Table 1.** *Cont.*

| Main Types of Ces | Components | Examples | Relative Impact Assessment | Possible Responses |
|---|---|---|---|---|
| Type 2 *Chronic* | - Chronic, low intensity of armed and fatal political violence<br>- Vulnerability to climate change-induced hazards<br>- High level of poverty<br>- Changing demographics between groups | **Sahel Region Mali** | - Large affected area<br>- Medium-to-high levels of displacement—internal, short-term, and circular<br>- Chronic food insecurity: collapse of markets and price hikes | **Continued presence in the region and food aid**<br>- Short-term distribution of food aid<br>- Aid to facilitate the resumption of agricultural activities<br>**Long-term measures**<br>- Aid for the long-term adaptation to climate change<br>- Plan for the integration of conflict parties |
| Type 3 *Urban* | - High level of civic violence: rioting and protests<br>- High level of exposure to climate change hazards<br>- High level of unemployment and high percentage of under-serviced population (public service)<br>- Unstable demographic dynamics: rural–urban migration and urban refugees | **Nairobi** (Kenya)<br>**Freetown** (Sierra Leone)<br>**Monrovia** (Liberia)<br>**Harare** (Zimbabwe) | - Localized affected area<br>- Epidemic outbreaks<br>- Concentrated forms of displacement<br>- Acute food insecurity: seasonal price hikes<br>- Large slum population | **Better service delivery to the population**<br>- Food aid<br>- Education<br>- Vaccination programs<br>- Cooperation over the reinforcement of health institutions<br>**Improve urban governance**<br>- Investment in urban employment<br>- Improved living standards for the poor |
| Type 4 *Protracted* | - Absence of central authority and large-scale protracted conflict with multiple non-state actors<br>- Severe vulnerability to climate change-induced hazards: consistently re-occurring and sudden disasters<br>- High level of poverty and collapse of state and local economies<br>- Disturbed demographics | **Somalia** | - Transnational with local hotspots<br>- Epidemic outbreaks<br>- Chronic food insecurity and famine: food availability<br>- Intermittent phases of displacement (e.g., Mogadishu) | **Reinstatement of central control and large-scale poverty reduction programs**<br>- Food aid distribution<br>- Investment for agricultural productivity<br>**Resumption of public services**<br>- Reinforcement of health institutions |

The year 2024 commemorates the 30th anniversary of the 1994 Rwandan genocide, one of the most severe and complex emergencies among all crises caused by the use of extreme violence against a civilian population. The Rwanda crisis caused a great shock in the field of humanitarian aid, prompting a profound reassessment of the international community's intervention strategies and its role in preventing and responding to crises [17]. Moreover, in the operational and technical sphere, it also marked the inception of a novel intervention approach, intertwining the ethical framework of actions with adherence to the humanitarian charter and minimum standards in humanitarian response and emphasis on outcome evaluation [18]. It seemed that a distinct "before and after" the Rwanda genocide would materialize, symbolizing a decisive step forward in enhancing humanitarian aid.

However, subsequent conflicts in regions like Sudan, Afghanistan, Ukraine, and Palestine indicate not only a lack of progress but also a discernible setback and regression, especially concerning adherence to the rules of international humanitarian law.

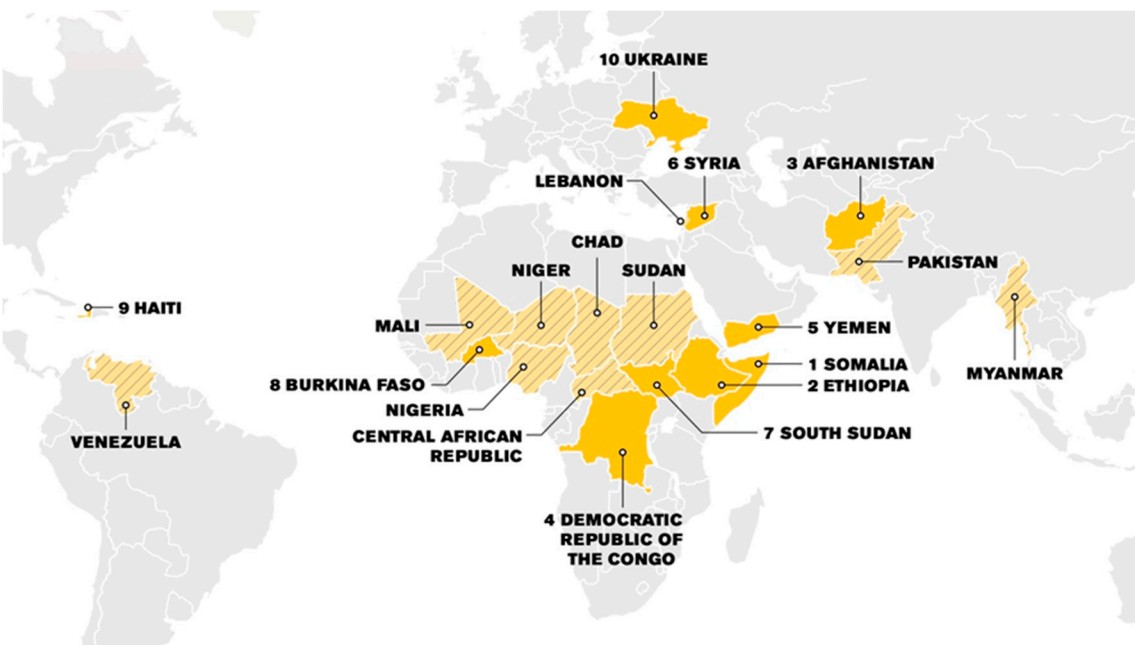

**Figure 1.** Map of ongoing complex emergencies of special relevance. The top ten ranked emergencies are in dark yellow color (source: International Rescue Committee. Emergency Wacht List 2023). Palestine had not yet been included at the time of the map's publication.

Viewed retrospectively, the global landscape of humanitarian aid has experienced a decline over recent decades, influenced by consecutive global economic downturns in 1973–1975, 1982, 1991, 2009, and 2020. These economic crises have impacted developmental levels and resulted in the degradation of international relations, marked by a near-disappearance of multilateralism as a model for inter-country operations amidst a shifting global geopolitical context. Compounding these challenges are the escalating climate crisis and a surge in extreme poverty and inequality levels, further exacerbated by the COVID-19 pandemic.

The current era is witnessing crises that are not only increasingly complex and protracted but also attract diminishing political interest for intervention, be it through humanitarian aid or essential activities in prevention, mitigation, or rehabilitation. According to the Global Humanitarian Assistance Report 2023, in the year 2022, a total of 44 countries experiencing prolonged crises accounted for 83% of the global necessity for aid. The call for humanitarian assistance reached a peak, with 406.6 million individuals across 82 countries in urgent need of support. Reflecting this surge, international humanitarian aid funding escalated to USD 31.3 billion in 2022, registering a growth of USD 800 million compared to 2020.

The world is also more fragile today, with greater differences between poor and rich countries. Globally, we are less equipped and coordinated to deal with multiple impacts caused by financial crises, natural hazard-related disasters, and violence, increasingly frequent phenomena that will likely spread beyond national borders and increase due to rapid demographic growth. If the current trends continue, by 2030, the cost of humanitarian assistance will have risen to USD 50 billion, and 62 percent of the world's poor will live in fragile and conflict-affected countries, a clear warning that humanitarian needs will increase even more [1]. Projections from the Intergovernmental Panel on Climate Change (IPCC) on the increasing intensity and frequency of climate-related disasters, as well as the

deterioration of peace indicators over the last decade, point in this direction, and the costs could be even higher than current estimates [19].

The traditional foundations and approaches of humanitarian aid appear insufficient in today's landscape of disasters and crises, which are increasingly complex and divergent, marked by a diminished capacity and shifting roles of various actors in alleviating suffering. Part of the solution to the problem involves rethinking the challenges and commitments involved in humanitarian aid. The following sections review the historical evolution of the conceptualization of humanitarian aid and address some of its ethical challenges and dilemmas.

## 3. The Evolution of the Concept and Practice of Humanitarian Aid

Traditionally, humanitarian aid has been understood as the action of providing basic survival assistance to communities affected by crises, conflicts, or disasters to alleviate suffering and recover a certain level of normality and carried out necessarily following humanitarian principles. Although the term *humanitarian action* is also used as synonymous with humanitarian aid, it differs in the fact that humanitarian action can also include other types of activities such as political lobbying, testimony, or public denunciation. We must also distinguish humanitarian aid from simple *emergency aid*. In this second case, actions do not necessarily have to adhere to and follow the classic principles of humanitarianism.

The evolution of humanitarian aid, from initial emergency aid or relief to humanitarian action, has gone through different phases with changes in approaches, strategies, and priorities since its founding phase in the 19th century by the Red Cross Movement and the Red Crescent, before then passing through a certain questioning of the neutrality and the need for testimony and denunciation proposed by Doctors Without Borders in the 1970s and 1980s.

The mid-1980s saw a far-reaching shift in the international aid system: a shift in donor policy from direct donor assistance to recognized governments to support private and non-governmental actors. In that decade, government funding for humanitarian aid increased significantly, and NGOs came to play an increasingly important role in providing that assistance. Since the 1980s, the reorganization of the humanitarian aid market has altered aid distribution patterns, shifting the focus from peripheral areas directly to the centers of conflicts. This change reflects a broader trend of confining the impacts of political crises within the unstable regions themselves. This development has raised important questions and problems for humanitarian aid organizations.

There are more evolving trends in the field of humanitarianism that can be considered. The first one is *Developmental Humanitarianism* [20,21] which emphasizes addressing the root causes of crises and working toward long-term solutions. It involves integrating humanitarian response with development initiatives to build resilience and sustainable solutions for communities affected by crises. That means combining the relief aid of the immediate response (focused on providing immediate assistance to address the basic needs of affected populations, such as food, water, shelter, and medical care, in the aftermath of a crisis) with a developmental approach or long-term planning (aimed at addressing the root causes of crises and building resilience in communities to reduce their vulnerability to future disasters or conflicts).

The second one is *Localized and Community-Centered Humanitarianism*. This trend emphasizes engaging local communities in humanitarian responses, recognizing the importance of understanding and incorporating local knowledge, capacities, and resources in shaping effective and culturally sensitive interventions. It also focuses on capacity building by empowering communities and strengthening the abilities of local governments and organizations to respond effectively to crises.

The third one is *Advocacy and Political Humanitarianism*, a more politically engaged form of humanitarianism that acknowledges the political dimensions of crises [22,23]. This involves advocacy for policy changes addressing root causes with a more explicit recognition of the political context. Humanitarian assistance can include actions to promote peace,

reconciliation, and conflict resolution, but advocacy and policy influence are necessary to raise awareness [24]. Humanitarian organizations must be engaged in advocacy efforts to raise awareness about the needs of affected populations and influence policies at local, national, and international levels to support humanitarian efforts better. This approach views humanitarian assistance through the lens of human rights, emphasizing the dignity and rights of affected populations.

From a much more critical perspective, there are approaches that even consider humanitarianism as a political, economic, and military interference in the internal affairs of a state justified by a transnational morality (transnational morality or rather transnational political realism?) characteristic of the period after the Cold War. This approach considers that humanitarianism has a simplistic worldview and that coercive humanitarian actions trigger negative consequences. It has been quite effective in protecting Western states from the collateral effects of political crises but less so in resolving the problems it claims to address, such as in the cases of Bosnia, Kosovo, Rwanda, and Darfur [25].

Regardless of whether aid is based on one or more of the mentioned approaches, there is a consensus among aid organizations that interventions must consider at least two essential elements. The first is to prioritize the needs of vulnerable and marginalized groups, such as women, children, the elderly, and people with disabilities, to ensure that aid is appropriate, relevant, and inclusive and meets the diverse needs of affected populations.

The second element is the environmental sustainability of the aid intervention to mitigate the environmental impact and work toward sustainable practices, as well as minimize negative effects on ecosystems and communities. These approaches are not mutually exclusive, and humanitarian organizations often employ a combination of strategies based on the specific context and needs of the affected population. Adaptability and collaboration are still key principles in effective humanitarian aid.

## 4. The Challenges and Ethical Dilemmas of Humanitarian Aid

Humanitarian aid faces challenges of a different nature; some of them have been present historically since its inception, and others have appeared as a consequence of the change in the current geopolitical scenario and the emergence of more complex and prolonged types of crises in which humanitarian intervention is also more difficult.

Today's crises are more dynamic, and the conditions under which they occur are changing rapidly. Therefore, some aid strategies may become obsolete, and organizations may have difficulty adapting to new situations and may not have sufficient or adequate intervention capacity due to them lacking the human resources, experience, or equipment necessary to respond effectively.

The first challenge of aid concerns *access restrictions* to the affected population. Humanitarian access [26] is the ability of humanitarian aid to reach the most vulnerable, and for the most vulnerable to reach humanitarian aid, and in the last few decades, there has been an escalation in the deliberate obstruction of access to humanitarian aid [27,28]. This challenge is not new, but it has been exacerbated by a growing perception of aid organizations as active elements in conflicts that can be used to increase visibility or managed for political purposes.

Denial of humanitarian access takes many forms, depending on the context [29]. In Afghanistan, the Taliban has banned the World Health Organization from working in crucial areas. In Yemen, severe movement constraints for humanitarian aid, aerial bombardments, and restrictions on lifesaving imports, including food, fuel, and medicine, have left millions teetering on the brink of famine. In northeast Nigeria, State armed forces coerce civilians into garrison towns in order to access emergency aid. In Syria, South Sudan, and Myanmar, governments and non-state actors unapologetically use siege, starvation, and obstruction as military and political tactics, putting millions of their own people at risk while impeding aid organizations from operating. More recently, Israel has deliberately prevented access to humanitarian aid in Palestine as a way to force displacement. Humanitarian access is

essential to protecting the rights, dignity, and safety of civilians affected by conflict, as established by international humanitarian law.

A second challenge concerns the ***risks to the security of interventions and humanitarian aid personnel***. Violence and insecurity are a reality for aid workers in conflicts [30]. Attacks on aid workers in 2022 claimed more lives (141 fatalities) than in any year since 2013. In addition to the 141 aid workers killed, 203 were wounded, and 117 were kidnapped. Since the beginning of 2023, 62 aid workers have been killed in violent attacks, a 40% increase from the same period in 2022 [31]. There are more attacks on aid workers worldwide, 90 percent of which targeted local aid workers. There have also been many attacks on vital physical infrastructure, like the attacks by the Israeli army on hospitals and ambulances in Palestine. South Sudan remains the most violent context for aid workers, followed by Afghanistan, Syria, Ethiopia, and Mali. Russia's invasion of Ukraine threatened to drive up the casualty numbers in 2022, with aid workers at risk of indiscriminate violence such as airstrikes, shelling, rocket attacks, and remnants of war. Insecurity limits the ability of aid organizations to deploy personnel and resources, affecting the delivery of assistance and its sustainability over time.

The third challenge is related to the ***lack or insufficiency of financing***. Humanitarian aid depends on funding from private donors, governments, and international organizations and, when inadequate, limits the scale and scope of aid efforts and the ability to address the needs of affected populations. An estimated 363 million people were in need of humanitarian aid in 2023, an increase of 37 million since the end of 2022. While the need has reached record levels, the gap between financial requirements and resources committed stands at its highest level ever: USD 41.4 billion for mid-2023 [32]. But in 2023, the United Nations Global Humanitarian Appeal was only 20% funded [33] and this is causing "a crisis within a crisis". Shortages of funds are causing rollbacks of food aid in Syria, Bangladesh, the Occupied Palestinian Territory, Afghanistan, and Yemen. The disparity between funding needs and funding received is driven by the changing nature of crises, and most of this aid is dedicated to civilians in regions struggling with protracted conflicts.

Political and bureaucratic barriers, such as governmental constraints on aid distribution, procedural delays in obtaining approvals, or restrictions imposed by legal frameworks, frequently hinder aid financing. This challenge transcends economic or financial dimensions and encompasses ethical considerations as well. It raises critical questions about the fairness and appropriateness of resource allocation, particularly whether the available aid resources are being directed to the population groups most in need.

To address this problem, it will be necessary to modify substantial aspects of humanitarian aid, several of which have a clear ethical dimension. Aid organizations and donors will need to improve financial transparency, provide more support and funding tools for national first responders, expand the use of cash-based programming, and improve coordination in its implementation. Also, to reduce duplication and management costs, regular reviews of functional spending, more joint and impartial needs assessments, and listening more to and including beneficiaries in the decisions that affect them. Lack of accountability is a problem with a strong ethical component. In some cases, there may be a lack of accountability within humanitarian organizations, leading to issues such as corruption, mismanagement of resources, or the diversion of aid away from its intended recipients.

A fourth challenge is related to the urgent ***need to restore respect for the basic elements of international humanitarian law (IHL)***. The humanitarian space has suffered a notable deterioration in recent years. Crises such as those in Ukraine and Palestine have shown the level of disregard for IHL shown by both Russia [34] and Israel [35] in their attacks on the civilian populations of Ukraine and Palestine, respectively. Denying access to water, food, or medical aid, forcing the widespread expulsion of the population, deliberately attacking civilians, and destroying hospitals or schools are war crimes that reveal the urgent need to recover respect for the protection elements established by IHL.

Several factors contribute to this trend of deterioration of humanitarian protection rights. The first one is that the nature of warfare has evolved, with an increase in non-state actors, asymmetric warfare, and the use of new technologies. These changes create challenges in applying traditional humanitarian law frameworks, making it difficult to regulate and enforce the rules effectively. Several ongoing armed conflicts often involve non-state actors and transnational groups that do not adhere to established international norms, as in the case of the Russian mercenary company Wagner, which acts in different conflicts in various geographical areas. In these cases, the parties involved do not recognize or respect IHL, leading to widespread violations.

The second factor is increasing impunity, with perpetrators of violations of IHL often left unpunished. Lack of accountability for war crimes and violations of human rights encourages a culture of impunity, where individuals, groups, and even states believe they can get away with such actions. The case of Israel systematically ignoring United Nations resolutions and recently brought to the International Criminal Court for its punishment of the civilian population of Palestine is paradigmatic of this impunity.

Other factors are related to large-scale migration and displacement resulting from conflicts that can strain resources and lead to social tensions in host countries, affecting their willingness or ability to uphold IHL. Also, misinformation, propaganda, and manipulation of public opinion through the media can shape how conflicts are perceived globally. This can sometimes lead to distorted views of the situation on the ground and impact efforts to address IHL violations. Addressing the deterioration of respect for IHL requires collective efforts from the international community, including stronger diplomatic initiatives, accountability mechanisms, and raising awareness about the importance of upholding humanitarian norms.

## 5. Conclusions

Humanitarian aid raises ethical dilemmas of a different nature that have worsened in recent decades. The reasons for this are the deterioration of the international economic and geopolitical context, international relations based on states' return to unilateralism and protectionism, and the loss of the capacity of multilateral organizations to guarantee respect for international humanitarian law.

These ethical dilemmas affect essential elements of humanitarian aid, such as an adequate selection of crises to which to provide aid and a selection of beneficiaries based on needs and not political or geostrategic criteria; neutrality against the aggressor or collaboration with governments that do not respect human rights; the allocation of resources and prioritization when they are limited; the safety and protection of aid recipients; cultural and political sensitivity and the recognition of local knowledge, skills, and capacities in responding to crises; the appropriateness, sustainability, and long-term impact of actions; security risks for aid personnel; transparency and accountability; the duty to report and civil activism in the face of the violation of human rights and the deterioration of respect for international humanitarian law; and the rights of affected groups and local communities in humanitarian decision-making and implementation.

As has been rightly pointed out by Phillipe Biberson and Francois Jean,

"Humanitarian aid has become the West's favored response to political crises that are not of major strategic importance. As such, it has become a foreign policy tool and a factor of legitimization of international intervention. This has brought with it the risk of humanitarian organizations becoming mere instruments in the hands of government authorities. At the same time, increasing intervention by governments in the humanitarian space—including armed intervention—has blurred the distinctions between the political and humanitarian rationale. Such confusion throws doubt on the independence and impartiality of humanitarian actors" [36].

**Funding:** This research was supported by the project "Deberes éticos en contextos de desastres (DESASTRE)", funded by the BBVA Foundation Grants for Scientific Research Projects 2021.

**Conflicts of Interest:** The authors declare no conflicts of interest.

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
