# Peer review of "The Evolution of Humanitarian Aid in Disasters: Ethical Implications and Future Challenges"

_philosophies, doi:10.3390/philosophies9030062_

Round 1

Reviewer 1 Report

Comments and Suggestions for Authors

Perhaps introduce the political dimension mentioned in the conclusion, earlier in the article.

Author Response

Dear Reviewer 1, we would like to thank you for your time and kind suggestions to improve our article. We completely agree with your recommendation and have revised it accordingly.

Reviewer 2 Report

Comments and Suggestions for Authors

This is a well written paper that describes and justifies the arguments the paper presents. The conclusion ties the paper together and provides a clarity to the discussion. All references were appropriate to the paper. While this is not a research paper, the paper adds to the body of knowledge on disaster by presenting clear information on the history and role of Humanitarian Aid in Disasters including the ethical stances and future challenges. Papers like this that pull together information and present them in a readable format are valuable to providing information on improving Disaster response and recovery.

Line 182 - review and rewrite - does this need to be in upper case and why the upside down ? at the beginning

Author Response

Dear reviewer 2, we would like to thank you for your time and kind suggestion to improve our article.
